# Identification of Genes Responsible for the Synthesis of Glycitein Isoflavones in Soybean Seeds

**DOI:** 10.3390/plants13020156

**Published:** 2024-01-05

**Authors:** Masaki Horitani, Risa Yamada, Kanami Taroura, Akari Maeda, Toyoaki Anai, Satoshi Watanabe

**Affiliations:** 1Faculty of Agriculture, Saga University, 1 Honjo-machi, Saga 840-8502, Japan; horitani@cc.saga-u.ac.jp (M.H.);; 2Faculty of Agriculture, Kyushu University, 744 Motooka, Nishi-ku, Fukuoka 819-0395, Japan; anai.toyoaki.494@m.kyushu-u.ac.jp

**Keywords:** *Glycine max*, isoflavone, QTL, glycitein, 6-hydroxydaidzein, flavonoid 6-hydroxylase, O-methyltransferase

## Abstract

Soybean (*Glycine max* (L.) Merrill) isoflavones are among the most important secondary metabolites, with functional benefits for human health. Soybeans accumulate three aglycone forms of isoflavones: genistein, daidzein, and glycitein. Soybean landrace Kumachi-1 does not accumulate malonylglycitin at all. Gene structure analysis indicated that *Glyma.11G108300* (*F6H4*) of Kumachi-1 has a 3.8-kbp insertion, resulting in a truncated flavonoid 6-hydroxylase (*F6H*) sequence compared to the wild-type sequence in Fukuyutaka. Mapping experiments using a mutant line (MUT1246) with a phenotype similar to that of Kumachi-1, with a single-nucleotide polymorphism (SNP) in *F6H4*, revealed co-segregation of this mutation and the absence of glycitein isoflavones. We also identified a mutant line (K01) that exhibited a change in the HPLC retention time of glycitein isoflavones, accumulating glycoside and malonylglycoside forms of 6-hydroxydaidzein. K01 contains an SNP that produces a premature stop codon in *Glyma.01G004200* (*IOMT3*), a novel soybean isoflavone O-methyltransferase (*IOMT*) gene. We further analyzed transgenic hairy roots of soybeans expressing *Glyma.11G108300* (*F6H4*) and *Glyma.01G004200* (*IOMT3*). Those overexpressing *F6H4* accumulated malonylglycoside forms of 6-hydroxydaidzein (M_6HD), and co-expression of *F6H4* and *IOMT3* increased the level of malonylglycitin but not of M_6HD. These results indicate that *F6H4* and *IOMT3* are responsible for glycitein biosynthesis in soybean seed hypocotyl.

## 1. Introduction

Soybean (*Glycine max* (L.) Merrill) is used as an oil and food source for humans and livestock. Soybean secondary metabolites have functional benefits to human health and have received much attention worldwide. These metabolites include isoflavones, which function in plant–microbe interactions [1,2] and have positive nutritional effects in the human diet. Three aglycone forms of isoflavones, genistein, daidzein, and glycitein, which are glycosylated and then malonylated, are synthesized in soybean mature seeds. These aglycones have health benefits, such as reducing the risk of cardiovascular disease and osteoporosis, mitigating menopausal symptoms [3], and preventing the proliferation of certain cancers [4].

The enzymes related to the isoflavones in legumes have been well elucidated (Appendix A), and their physical interactions among key enzymes have been studied [5]. Chalcone synthase (CHS) synthesizes naringenin chalcone from 4-coumaroyl-CoA and three units of malonyl-CoA. CHS, and chalcone reductase (CHR) also catalyze isoliquiritigenin synthesis [6,7]. Chalcone isomerase converts both chalcones (naringenin chalcone and isoliquiritigenin) into isoflavone aglycones (genistein and daidzein) with chalcone isomerase [5], isoflavone synthase (IFS [8,9]), and 2-hydroxyisoflavanone dehydratase (HID) sequentially [10]. Glycitein aglycone is synthesized from liquiritigenin by flavonoid 6-hydroxylase (F6H, a member of the cytochrome P450 family [11]), IFS, HID, and *O*-methyltransferase (OMT [12]). These aglycones are synthesized on the surface of the endoplasmic reticulum, and glycosyltransferases and malonyltransferases then catalyze the synthesis of glycosides (daidzin, glycitin, and genistin) and malonylglycosides (malonyldaidzin, malonylglycitin, and malonylgenistin) [13,14]. Many QTL studies on isoflavone content have been conducted [15,16,17,18,19,20,21,22,23,24,25,26,27], and the soybase database (https://www.soybase.org/, accessed on 1 October 2023) lists at least 289 QTLs related to isoflavones. However, knowledge of the genes responsible for the genetic diversity of isoflavone content in soybean genetic diversity is still limited. Several genome-wide association study (GWAS) analyses of isoflavone content have been performed [28,29]. Although an association has been found between isoflavone content and single-nucleotide polymorphisms (SNPs) in IFS genes, detailed effects of these SNPs have not been elucidated [30]. GWAS analysis has been performed with over 6 million SNP data points and 1500 germplasm accessions mainly obtained from China to identify the association between SNPs and total isoflavone content or malonylglycitin content, resulting in the identification of several SNPs significantly associated with malonylglycitin content and one major QTL on chromosome 11 (Gm11) [28]. They found GWAS peaks in the region 8.25 to 8.26 on Gm11 under four environmental conditions and proposed one candidate gene (*Glyma.11G108100*, 8.23 Mbp of Gm11). However, there have been no additional experiments, such as transformation and screening of mutant lines, to reveal the function of their candidate genes. We have also identified a major QTL for malonylglycitein content, *qMgly-11*, in the same region [31]. Three types of alleles—activated type (Aokimame (AOKI) allele), normal type (Fukuyutaka (FUKU) allele), and null type (Kumachi-1 allele)—were characterized at this locus [31]. This QTL regulates the accumulation of glycitein isoflavones in the hypocotyl of soybean seeds. We performed the QTL analysis to find *qMgly-11* with two different segregated populations. One population was obtained from a cross between FUKU and AOKI and another was obtained from a cross between FUKU and Kumachi-1. The closest DNA markers positions for *qMgly-11* were in the 8.18 to 8.21 Mbp region of Gm11 (Appendix A). Complete co-segregation between marker genotypes and the null allele or accumulation of glyciten isoflavones phenotypes was observed [31]. We also performed a GWAS analysis of 158 soybean landrace lines. The strongest association was detected with a DNA marker located at 8.16 Mbp. These results showed that the same chromosomal region (*qMgly-11* locus: 8.15 to 8.25 Mbp on Gm11) has significant association with the malonylglycitein content in soybean seeds [31]. Several genes and enzymes involved in glycitein isoflavone biosynthesis have been characterized [12,32], including three *F6H* genes (*F6H1* to *F6H3*) and two *IOMT* genes (*GmIOMT1* and *GmIOMT2*), but the relationship between *qMgly-11* and these genes had not been elucidated until now. The *qMgly-11* locus (8.15 to 8.25 Mbp) include 17 genes (*Glyma.11G107200* to *Glyma.11G109000*) in the soybean genome database (Wm82.a4.v1). We examined the annotation information of these genes and Glyma.11G108300 (8.25 Mbp on Gm11), annotated as p450, showed homology with other F6H proteins. We considered this gene a reasonable a candidate of *qMgly-11*. In this study, we identified a mutant line (MUT1246) showing the same phenotype as Kumachi-1, with a null allele for glycitein isoflavones. We identified *Glyma.11G108300* as a novel *F6H* gene (*F6H4*) underlying this QTL via gene structure analysis and mapping experiments. We further identified a novel mutant line, K01, showing peak shifts in high-pressure liquid chromatography (HPLC) corresponding to decreases in glycitin and malonylglycitin and increases in glycoside and malonylglycoside forms of 6-hydroxydaidzein. Mapping experiments revealed that a nonsense mutation in a novel *IOMT* gene (*IOMT3*) is associated with this peak-shift phenotype.

## 2. Results

### 2.1. Gene Structure of Soybean Landrace Kumachi-1 F6H4 Gene

To determine the gene structure of *Glyma.11G108300* (*F6H4*) of Kumachi-1, PCR amplification was attempted from the start codon to the stop codon. We could amplify the start of the second exon from FUKU but not from Kumachi-1 (Figure 1); therefore, we expected an insertion of DNA in *F6H4* from Kumachi-1. The combination of long-read sequences from Nanopore and short reads of Illumina sequencing data revealed that a 3.8 kbp insertion was present in Kumachi-1 *F6H4*. The inserted sequence was highly similar to the sequence of the retrotransposon *Tnt1* and had 670 bp terminal repeat sequences, which we could detect with PCR (Figure 1). The predicted amino acid sequence of Kumachi-1 F6H4 was truncated (80 aa) compared to the FUKU wild-type amino acid sequence (507 aa). This result indicates that Kumachi-1 has a null allele of *F6H4* and cannot synthesize glycitein isoflavones in its seeds.

### 2.2. Novel Mutant Line for F6H4 Gene

We then looked for additional mutant lines in the soybean mutant library that had no glycitein isoflavones and identified a mutant line (MUT1246) with a phenotype similar to that of Kumachi-1 (Figure 2A). Direct sequencing identified an SNP (G407A) in F6H4 of MUT1246 (Figure 2B), which caused a non-synonymous substitution in the amino acid sequence (Cys136Tyr). We confirmed the co-segregation of this mutation and the absence of glycitein isoflavones (glycitin and malonylglycitin) in the hypocotyls of a segregating population obtained from a cross between MUT1246 and TOYO (Figure 2C, Appendix A). Co-segregation in two mapping populations, one obtained from a cross between Kumachi-1 and FUKU in a previous study [31] and the other obtained from a cross between MUT1246 and TOYO in this study, indicated that these mutations in F6H4 resulted in the absence of glycitein isoflavone in soybean seed hypocotyl parts.

### 2.3. Association Study with Soybean Mini-Core Collection

We previously found a QTL for glycitein isoflavone content in a population obtained from a cross between AOKI and FUKU [31]. The AOKI allele of *qMgly-11* can increase glycitein content compared to that produced by the FUKU allele. If we assume that the gene underlying *qMgly-11* is *F6H4* (*Glyma.11G108300*), there is an explanation for the functional difference between the AOKI and FUKU alleles. These two lines had amino acid substitutions in F6H4 (A66T, T127I, A203S, and T248A, caused by SNPs G196A, C410T, G607T, and A742G, respectively). We performed association mapping with the soybean mini-core collection [33] to identify the amino acid substitutions responsible for the functional difference between the two alleles and found a significant association of A203S and T248A with glycitein isoflavone content (Appendix A). Linear regression analysis with both alleles in a single model that considered their interaction revealed that T248A had a significant association with the phenotype (*p* < 0.001) but that A203S did not (*t*-value = 0.18, *p* = 0.86, Figure 3). Mapping experiments with MUT1246 and association mapping using the soybean mini-core collection revealed that the C136Y and T248A substitutions affected the enzymatic activity of F6H4.

### 2.4. Model of F6H4 Protein Structure

The three-dimensional structure of F6H4 (AOKI type) with heme was modeled as mentioned in the materials and methods. The overall predicted structure is shown in Figure 4. Most plant cytochrome P450s are expressed on endoplasmic reticulum and are known as microsomal P450s; they have a transmembrane anchor at the N-terminus attached to the endoplasmic reticulum. The predicted structure of F6H4 had a long α-helix at the N-terminus, suggesting an anchor to the endoplasmic reticulum. The top view showed a triangular overall structure, which is characteristic of cytochrome P450s. The heme iron was co-ordinated by Cys446, and one propionate of the heme was located at a hydrogen-bond distance from arginine 444, which is conserved in all cytochrome P450s. These observations confirmed that the proposed structure was plausible. Next, we attempted to calculate the substrate-binding structures. Unfortunately, no reasonable structures with substrates were established by docking simulations. The structure built by AlphaFill can predict the position of not only heme but also of substrates registered in the protein data bank. We obtained a model structure of F6H4 with 4-phenyl-1H-imidazole (4PI), which was used as a substrate analog in some P450 crystallographic studies [34,35]. The distance between the hydroxylated carbon of the substrate and the iron ion in the heme is less than 4.5 Å, according to reported crystal structures of P450s [36]. Thus, the substrate for F6H4, liquiritigenin, was placed with this constraint based on the model structure with 4PI (Figure 4 and Appendix A). Importantly, the hydroxyl (4′-OH) on the side opposite the hydroxylation in liquiritigenin was located at a hydrogen-bond distance of 3.5 Å from threonine 248 in the Aokimame type, which was mutated to alanine in the FUKU type (Figure 4A). In addition, our predicted model revealed that cysteine 136 was located near the heme and formed a part of the heme pocket. Therefore, the C136Y and T248A mutations may be related to F6H4 enzymatic activity.

### 2.5. Expression Levels of F6H4 during Soybean Seed Development

We compared the glycitein isoflavone accumulation pattern and expression profile of *Glyma.11G108300* (*F6H4*) in developing soybean seeds (Figure 5A). Glycitein isoflavone levels were very low in the cotyledons, and the expression of *F6H4* in the cotyledons was also low at all sampling stages (I to VII; Figure 5B,C). In contrast, glycitein isoflavones accumulated in the hypocotyl (here, including the plumule, epicotyl, hypocotyl, and radicle) of soybean seeds during stages II to VI. The expression pattern of *F6H4* corresponded to the accumulation pattern of glycitein isoflavones. The highest level of glycitein isoflavone appeared at stage VI, two stages (>10 days) after the peak of gene expression of *F6H4*. The expression of *F6H4* increased, starting at stage II, peaked at stage IV, and then decreased until stage VII (Figure 5C). The association between the accumulation pattern of glycitein isoflavones and the expression profile of *F6H4* suggests that this gene is involved in glycitein isoflavone synthesis.

### 2.6. Phylogenetic Analysis of F6H Amino Acids Sequences

Several *F6H* genes (*F6H1*: *Glyma.18G080400* in Wm82.a1.v2; *F6H2*: *Glyma.18G080200*; and *F6H3*: *Glyma.08G326900*) have been characterized [32]. We compared the predicted amino acid sequences of F6H4 and these other soybean F6H proteins. A phylogenetic tree of homologs from soybean, *Lotus japonicus*, and *Medicago truncatula* F6H amino acid sequences showed F6H1 to F6H3 in one clade and F6H4 in another (Figure 6). The differentiation of two types of F6H proteins appeared to be conserved in legume plant species (Figure 6). The amino acid sequence identity and similarity between the predicted product of Glyma.11G108300 and that of the well-characterized F6H1 amino acids sequence (Glyma.18G080400) were 33% (168/505 aa) and 52% (264/505 aa). Within the soybean genome, *F6H1* has at least two homologs, *F6H2* and *F6H3*, and *Glyma.11G108300* (*F6H4*) also has at least two homologs, *Glyma.10G203500* and *Glyma.20G14800*. These results indicate that at least two genes related to glycitein isoflavone biosynthesis evolved independently.

### 2.7. Identification of Novel Mutant Line for Glycitein Isoflavones

During mutant line screening, we also identified a unique mutant line (K01) that showed a change in the HPLC retention time of glycitein isoflavones. This line showed a shift in the peak positions on the chromatogram: specifically, the glycitin and malonylglycitin peaks were replaced by peaks with an earlier retention time compared with the wild type (Figure 7). We mapped the mutated gene in K01 by using a population obtained from a cross between TOYO and K01. Selective genotyping analysis identified several DNA markers, located in the 1.0 to 4.0 Mbp region on Gm01, that were linked to the mutant phenotype. Further QTL analysis with all individuals in the population identified a QTL peak at the 0.43 Mbp position of Gm01 (Figure 8A). The marker genotypes at this position (13cM at P1_430457) coincided with the glycitein accumulation pattern in hypocotyl, and mutant (FUKU genetic background) homozygous plants showed alterations in content from high malonylglycitin to high malonylglycoside forms of 6-hydroxydaidzein (Figure 8B, Appendix A). Glycitein biosynthesis from liquiritigenin is catalyzed by F6H, IFS, and IOMT via 6-hydroxyliquiritigenin and 6-hydroxydaidzein (Appendix A). Among these enzymes, IFS is essential for the synthesis of genistein and daidzein, and K01 shows no difference in genistein- (genistin and malonylgenistin) and daidzein- (daidzin and malonyldaidzin) related isoflavones content compared to FUKU. Therefore, we assumed that a mutation in *IOMT* was a candidate for the K01 mutated phenotype.

A survey of SNPs between K01 and FUKU with whole-genome NGS analysis revealed 16 SNPs in the 1.0 to 3.0 Mbp region on Gm01. Among these SNPs, an allele of Gm01_430457 caused a stop codon mutation in Glyma.01G004200 (238 aa; Figure 8C), annotated as methyltransferase in the soybean genome database (Wm82.a2.v1). In K01, the guanine nucleotide at position 554 was changed to adenine, changing the codon for tryptophan (TGG) to a stop codon (TAG) at the 185-aa position. We also analyzed the molecular weight of the K01-type aglycone after acid hydrolysis of isoflavones extracted from K01 seeds. The retention time of the K01-type aglycone coincided with that of 6-hydroxydaidzein (6-HD, Appendix A), and mass-to-charge ratios (*m*/*z*) of glycoside and malonylglycoside forms of 6-hydroxydaidzein were identical to those of the shifted peaks in K01 (Appendix A). These results—a nonsense mutation in the candidate gene (*IOMT*) of K01 and 6-HD isoflavone accumulation in K01—indicate that the product of a novel *IOMT* (*Glyma.01G004200*) gene, hereafter designated *IOMT3*, is responsible for the conversion of 6-HD to glycitein. The expression profile of *IOMT3* overlapped with that of *F6H4*. High expression was detected in hypocotyl, with the highest level at stage IV (Appendix A), as in *F6H4* (Figure 5C). Plant OMT genes are categorized into two main groups: cation-independent (type I) and cation-dependent (type II) [37]. The amino acid sequences of representative OMT proteins [12] were compared to those of IOMT3 and its soybean homologs (Figure 9). IOMT3 was grouped into the type II OMT clade. The amino acid sequence identity and similarity between GmIOMT1 and IOMT3 were 126/228 (55%) and 162/228 (71%), respectively.

### 2.8. Transgenic Hairy Roots Expressing F6H4 and IOMT3

We further analyzed transgenic hairy roots of soybeans expressing *F6H4* only, *IOMT3* only, and *F6H4* and *IOMT3* together, all under the control of the ubiquitin promoter (Appendix A and Figure 10). The transgenic hairy roots overexpressing *F6H4* only showed accumulation of malonylglycoside forms of 6-hydroxydaidzein (M_6HD) and a seven-fold (*p* < 0.001) increase in total glycitein-related isoflavones (glycitin, malonylglycitin, and M_6HD) compared to those transformed with the empty vector. However, overexpression of *IOMT3* barely affected the isoflavone accumulation pattern compared to the empty vector. Co-expression of *F6H4* and *IOMT3* significantly increased the level of glycitin and malonylglycitin but not that of M_6HD. In addition, the total amount of isoflavone was also increased by *F6H4* and *IOMT3* co-expression.

All results shown in this study indicate that *F6H4* and *IOMT3* are responsible for glycitein biosynthesis in soybean seeds, especially within the “hypocotyl” (i.e., non-cotyledon seed tissues).

## 3. Discussion

### 3.1. Use of Mutant-Based Screening for Gene Identification

In this study, we identified two genes responsible for glycitein biosynthesis from two mutant lines, one showing the absence of glycitein isoflavones and the other showing changed peak positions on an HPLC chromatogram. The mutant lines with such phenotypes have so far not been the subject of any reports. One responsible gene is *F6H4* (*Glyma.11G108300*) and the other is *IOMT3* (*Glyma.01G004200*). The location of *Glyma.11G108300* coincided well with that of the *qMgly-11* region identified in previous studies [37], and the effect of *qMgly-11* explained 56.5% of the phenotypic diversity of seed malonylglycitein content in soybean mini-core collections. In addition, GWAS analysis of over 1500 soybean germplasms with four repeated evaluations for malonyglycitin content identified the highest peak at the same positions [28]. The results of these studies indicate that *qMgly-11* is a major and stable QTL and would be a valuable locus for soybean breeders trying to improve soybean seed isoflavone components. Elicitor-inducible glycitein biosynthetic genes have been identified and characterized with transcript analysis using a inoculated or elicitor-treated soybean seedlings [11,12]; however, reports related to mutant-based screening of the isoflavone content in soybean seeds are limited. Our previous study identified a *GmCHR5* mutant that showed a decrease in daidzein-related isoflavones compared with genistein derivatives [38]. Some authors have conducted GWAS using many germplasm accessions and extensive SNP information [28,29]. However, the identification of genes responsible for regulating isoflavone content and its components remains limited in these studies. These results indicate that the wide genetic diversity used in GWAS analysis tends to mask the effects of single genes with small effects and the effect of rare alleles, and it decreases the power to detect multiple alleles in germplasm collections [39]. Mutagenesis can also increase genetic diversity related to the regulation of isoflavone content and is useful for further soybean breeding programs that focus on the improvement in isoflavone components. Therefore, screening mutant lines for alterations in isoflavone content and components and identifying the genes responsible for these phenotypes are important for revealing the genetic pathways related to the synthesis and accumulation of isoflavones and the expansion of soybean genetic diversity for these traits.

### 3.2. Several Independent Pathways of Glycitein Biosynthesis in Soybeans

A previous study [12] identified a unique type of *IOMT*, *GmIOMT1 (Glyma.05G147000)*, from a fungus-inoculated soybean seedling by using metabolome and transcriptome analysis based on the “guilt-by-association” principle [12]. Analysis of transgenic hairy roots overexpressing F6H1 (Glyma.18G080400) and GmIOMT1 showed accumulation of glycitein isoflavones [12]. Phylogenetic analysis based on their amino acid sequences indicated that the differentiation between F6H1 (Glyma.18G080400) and F6H4 (Figure 6) and between GmIOMT1 (Glyma.05G147000) and IOMT3 (Figure 9) did not originate from genome duplication during soybean evolution because, in each case, this differentiation came earlier in legume evolution. Given their evolutionary relationships, it is likely that there are two independent pathways related to isoflavone synthesis in soybean and other legumes: one triggered by inoculation with fungi and the other functioning during seed development. These pathways probably share the same *CHS* and *IFS* genes for genistein synthesis. However, glycitein synthesis is controlled by different pathways, depending on the situation. In some cases, daidzein functions as a precursor for the phytoalexin glyceollin [40]. Induction of glyceollin through fungal inoculation also induced glycitein isoflavone synthesis [12]. One possibility is that the accumulation of glycitein isoflavone influences the amount of glyceollin in plant tissue. Further studies are necessary to determine the physiological significance of glycitein accumulation. In contrast, developing soybean seeds also accumulate isoflavones in their cotyledons and hypocotyls. The isoflavone accumulation patterns in these two tissues are also probably controlled by different mechanisms [31]. The correlation between the amount of glycosides and that of malonylglycosides of daidzin, genistin, and glycitein was very high (0.97 to 0.98) in the cotyledon and 0.89 to 0.96 in the hypocotyl within a wide sample of soybean varieties [38]. In contrast, the correlations between the amount of glycoside or malonylglycoside isoflavones between the cotyledon and hypocotyl were not significant (0.02 to 0.09 for glycoside and 0.03 to 0.14 for malonylglycoside). These phenomena indicate that the biosynthesis pathways of isoflavones in soybean are controlled by different mechanisms that depend on the type of plant tissue (root, leaves, cotyledon, and hypocotyl) and biotic (e.g., stimuli from the elicitor) or abiotic stresses, such as irrigation [41] or temperature during seed-filling periods [42]. Soybean roots also produce the aglycone form of isoflavones from isoflavones accumulated in vacuoles via demodification, such as deglycosylation [43]. There are no data yet on isoflavone content for the mutant lines (MUT1246 and K01) at the whole-plant level, such as in roots or leaves, but it is likely that nodulation and other plant growth characteristics could be affected by mutations in *F6H4* and *IOMT3*. Thus, further studies using these mutant lines will be useful for understanding the functional effects of these genes at the whole-plant level.

### 3.3. Effects of Enzyme Structure on Glycitein Biosynthesis

Mapping experiments with MUT1246 and association mapping of the soybean mini-core collection revealed that the C136Y mutation eliminated glycitein production and that the T248A mutation reduced it. The amino acid position 136 formed a heme-binding pocket (Appendix A). Thus, we propose that the mutation of cysteine 136 to tyrosine affects the heme-binding mode and induces the loss of mono-oxygenase activity because the side chain of tyrosine is quite bulky compared to that of cysteine (Appendix A). In contrast, the T248A mutation preserved enzymatic activity. According to our predicted three-dimensional structure with liquiritigenin, the hydroxyl at its 4′ position is located at a hydrogen-bond distance from the threonine side chain. This suggests that this bond plays an important role in retaining the substrate in the precise position and the proximity of carbon 6C of liquiritigenin to the heme iron (iron–oxo or iron–peroxo intermediates). Meanwhile, the T248A mutation preserves the capacity for substrate binding in the active site pocket, but the substrate would be perturbed, and this mutation reduces the activity because the side chain of alanine has less volume. Thus, we conclude that threonine 248 is important for the precise formation of a ternary complex (heme–liquiritigenin–enzyme).

### 3.4. The Possibility to Produce Novel Materials for Human Dietary Nutrition

Isoflavones contribute to the taste of soybean seeds, particularly astringent tastes [44]. Among the three isoflavone aglycones, the bitterness order is glycitein > daidzein > genistein. Thus, the glycitein-absent null allele found in MUT1246 could become a good source for developing new soybean varieties with improved taste. In future, sensory testing using processed products made from soybean seeds defective in glycitein isoflavones will be necessary to evaluate whether the absence of glycitein has a positive effect on the taste of processed foods such as soymilk, tofu, or natto.

As discussed above, the K01 mutant line accumulates the malonylglycoside forms of 6-hydroxydaidzein. Our preliminary screening using the Japanese and world soybean mini-core collection [33] identified no soybean accessions with the same phenotype as K01. Glycitein metabolism was analyzed by using rat and human liver microsomes, and human fecal flora were investigated [45]. In that study, 6-HD was detected among the metabolites obtained from glycitein digestion. In addition, 6-HD has antioxidant activity in cell culture models, and the antioxidant activity of dietary isoflavones may be due to the formation of antioxidant metabolites of 6-HD [46]. These results indicate that 6-HD has a nutritional function in the human diet. The combination of the AOKI-type *qMgly-11* allele (threonine-248 type of F6H4) increasing F6H4 activity and the K01 mutant allele causing the accumulation of 6-HD malonylglycoside offers the possibility to develop “super” 6-HD-accumulating soybean seeds with high antioxidant effects. By stacking these alleles in future breeding programs, new soybean materials with novel nutritional features could be developed.

## 4. Materials and Methods

### 4.1. Mutant Library

We screened our previously reported soybean mutant library in the Fukuyutaka (FUKU) genetic background [47] for lines showing drastically reduced or absent malonylglycitin. Among 2831 mutant lines, we found that the MUT1246 and K01 lines displayed different malonylglycitin contents than wild-type FUKU. We propagated the seeds of these mutant lines and used them as parental lines for mapping experiments.

### 4.2. Plant Materials

The two mutant lines were crossed with Japanese soybean cultivar Toyoshirome (TOYO) (National Agriculture and Food Research Organization [NARO] Genebank; bred at the Kyushu Okinawa Agricultural Research Center, NARO), which has a different genetic background than FUKU. The F_2_ population obtained from a cross between MUT1246 and TOYO, containing 96 individuals, was used to confirm co-segregation of functional SNPs and their phenotypes. Another F_2_ population containing 96 plants was obtained from a cross between K01 and TOYO and used for mapping the mutated gene in the K01 line. The F_2_ seeds from each cross were used for HPLC and DNA analyses.

### 4.3. NGS Analysis

DNA used for whole-genome sequencing analysis was extracted from FUKU, Kumachi-1, and the mutant line K01. Next-generation sequencing (NGS) analysis was outsourced to Novogene (Beijing, China). Sequences approximately 10 to 13 Gbp in length were obtained for each sample. Long-read sequence data from Kumachi-1 were also obtained using a Minion sequencer (Oxford Nanopore Technologies, Oxford, UK) according to the manufacturer’s instructions. A soybean reference sequence (Wm82.a2.v1) was obtained from a public database (https://phytozome-next.jgi.doe.gov/, accessed on 1 April 2021). To map the mutated gene in K01, FASTQ files containing sequence data were used to align short reads to a candidate region (Gm01 1.0 to 3.0 Mbp) in the reference sequence using the Bowtie 2 program (ver. 2.3.5.1 [48]). The SAMtools suite (ver. 0.1.18 [49]) was used to detect polymorphisms between FUKU and K01. We evaluated the functions of the detected SNPs by using the snpEFF program [50]. We also developed DNA markers to detect K01 and MUT1246 functional nucleotide polymorphisms using the nearest neighboring nucleotide substitution high-resolution melting (NNNs-HRM) technique and designed primers to detect SNPs, according to a previous study [51]. Primer sequences used in this study are summarized in Appendix A.

### 4.4. Mapping Experiments

DNA was extracted from powdered F_2_ seeds using an automated DNA extraction machine (GENE PREP STAR PI-480; Kurabo Industries Ltd., Osaka, Japan), according to the manufacturer’s instructions. DNA markers based on NNNS-HRM that detected polymorphisms between FUKU (the genetic background of MUT1246 and K01) and TOYO were used for the mapping experiments, according to a previous study [38]. PCR experiments for all marker genotyping were conducted in a quantitative thermal cycler (LightCycler 96 System; Roche Diagnostics K.K., Minato, Tokyo, Japan). DNA marker information for mapping the K01 mutated locus and PCR conditions was provided in a previous study [51]. A selective genotyping technique [52] was used to identify candidate loci for the K01 mutated locus, with 10 individuals showing the K01-type isoflavone phenotype. Additional DNA markers for SNPs unique to each mutant line are summarized in Appendix A.

### 4.5. Construct Preparation for Ectopic Expression of Candidate Genes and Soybean Hairy Root Transformation

Full-length coding sequences of *F6H4* and *IOMT3* were amplified from AOKI and FUKU developing seed cDNA, respectively, by PCR using KOD FX Neo polymerase (Toyobo, Osaka, Japan). Each PCR product was cloned into the pENTER/D-TOPO or pDONR221 entry vector (Thermo Fisher Scientific, Waltham, MA, USA) and sequenced. The cloned fragments were transferred into the destination vector pUB-GW-GFP [53] using the Gateway LR recombination reaction (Thermo Fisher Scientific). The resulting vectors were designated as the control (pUB-GW-GFP empty vector), F6H4 (containing Glyma.11G108300), and IOMT3 (containing Glyma.01G004200).

To produce a vector co-expressing *F6H4* and *IOMT3* (designated F6H4-IOMT3)*,* the DNA sequence from the promoter to the terminator of *F6H4* was amplified from the F6H4 vector using primers created for In-Fusion Cloning technology (Takara Bio, Kusatsu, Japan). The destination vector containing *IOMT* was digested with *Sac*I, and a PCR fragment containing the promoter to terminator of *F6H4* was inserted upstream of *IOMT3* by homologous recombination. The primers used for cloning all of the genes are listed in Appendix A.

All of the newly constructed vectors were introduced into the *Agrobacterium rhizogenes* strain LBA1334 using the freeze–thaw method [54].

Transgenic hairy roots were generated as previously described [55] with slight modifications [38]. Briefly, FUKU seeds were infected with LBA1334 harboring the above vectors, and transgenic hairy roots with strong green fluorescent protein (GFP) signals were selected, weighed, separated into two 3 mL plastic tubes with 5 mm metal beads (Yasuikikai, Osaka, Japan), frozen in liquid nitrogen, crushed with a Multi-Beads Shocker (Yasuikikai), and used for RNA extraction and isoflavone measurement.

### 4.6. Measurement of Isoflavone Content

We extracted isoflavones from cotyledons, embryo tissues, and transgenic hairy roots. For analysis of the segregating populations, we removed the seed coat and separated seeds into cotyledons and “hypocotyls” (here, consisting of plumule, epicotyl, hypocotyl, and radicle) with a knife. Freeze-dried hypocotyl samples were weighed and then crushed in a Multi-Beads Shocker in a 3 mL tube with metal beads. Seed powder (100 mg) obtained from cotyledons or crushed powder of hypocotyls was suspended in 1 mL of extraction buffer [70% ethanol (*v*/*v*) and 0.1% acetic acid (*v*/*v*)]. The mixture was vortexed and sonicated (30 min at room temperature) using an ultrasonic cleaner (Bransonic 5800; Emerson Japan Ltd., Kanagawa, Japan). The mixture was briefly centrifuged, and the supernatant was transferred to a new tube. These extraction steps were repeated three times, and the total extract (approximately 3 mL) was filtered (0.45 µm membrane). Samples (3 µL) were analyzed by HPLC (Jasco Corp., Tokyo, Japan) on a Hydrosphere C18 column (YMC Co., Ltd., Kyoto, Japan). The details of the separation conditions (water-based two-solvent systems) have been described previously [31]. Isoflavones were detected at 254 nm using an ultraviolet (UV) detector.

Standard curves for daidzein, daidzin, malonyldaidzin, glycitein, glycitin, malonylglycitin, genistein, genistin, malonylgenistin (Wako, Osaka, Japan), and 6-hydroxydaidzein (Tokyo Chemical Industry Co., Ltd., Tokyo, Japan) were constructed. The retention times and areas of the isoflavone standards were used to calculate the amounts of isoflavones from the peak areas of the samples.

The mass-to-charge ratios (*m*/*z*) of the K01-type isoflavones were measured using LCMS-2020 (SHIMADZU CORPORATION, Kyoto, Japan), which gave *m*/*z* 417 for daidzin, 433 for genistin and the glycosylated form of 6-hydroxydaidzein, and 447 for glycitein. The HPLC conditions were as described above.

Acid hydrolysis of isoflavones was performed as follows: 10 hypocotyls obtained from K01 mutant seeds were crushed using a Multi-Beads Shocker. An isoflavone extract (3 mL) was prepared as described above. A total of 1 mL of this solution was mixed with 1 mL of 70% EtOH in HCl solution and heated for 2 h at 95 °C. NaOH solution (1.5 M) was added for neutralization (pH7 to 8). Litmus paper was used to determine the pH of the solutions. After neutralization, 2 mL of ethyl acetate was added, and the solution was vortexed and centrifuged briefly. The supernatant (water phase) was removed, and the ethyl acetate phase was transferred into a new test tube and evaporated to dryness in a heating block (80 °C). Finally, the isoflavone hydrolysates were dissolved in 1 mL of extraction buffer (described above) and analyzed by HPLC.

### 4.7. Expression Analysis

The parental line of FUKU was grown in the field of Saga University (Honjo campus; lat 33.242, long 130.288; June to November 2019) under natural day-length conditions. Developing seeds were collected randomly from several plants and the developmental status of each stage was as follows: stage I, seeds with indistinguishable hypocotyl and cotyledons (~3 mm); II, seed length size 3–5 mm; III, seed size 5–7 mm; IV, seed size 7–8 mm; V, 40 days after flowering (DAF40), late seed enlargement stage, seed size over 8 mm; VI, 50 days after flowering (DAF50); and VII, 60 days after flowering (DAF60). Total RNA was isolated from the tissues (50–100 mg) using the Total RNA Extraction Kit Mini Plant (SciTrove, Tokyo, Japan) with additional rDNase I (Takara Bio Inc., Shiga, Japan) treatment. cDNA was synthesized from 1 µg total RNA using ReverTra Ace (Toyobo, Osaka, Japan). A 5 µL aliquot (approximately 25 ng) of cDNA was used as a template for PCR. Quantitative real-time PCR was conducted as follows in a LightCycler 96 System: 95 °C for 5 min for enzyme activation, followed by 95 °C for 15 s, 60 °C for 15 s, and 72 °C for 20 s for a total of 40 amplification cycles. EvaGreen Dye (20× in water; Biotium, Inc., Fremont, CA, USA) was used as the fluorescent dye, and dNTPs (Takara), homemade recombinant *Taq* polymerase, the PCR buffer mentioned in a previous study [51], and 1 pmol of each gene-specific primer were used to evaluate transcript levels of target genes. Expression relative to the internal reference gene, *GmELFa* (*Glyma.19G052400*), was calculated using the 2^−ΔΔCt^ method [56]. Data from three or four biological replicates were analyzed. Primers for expression analysis are listed in Appendix A. 

### 4.8. Protein Structure Modeling

With nearly 40,000 crystal structures of cytochrome P450s reported to date, including more than 400 crystal structures of F6Hs, the predicted structures of P450s are highly reliable. The structure of F6H4 (AOKI type) was modeled by using AlphaFold2_mmseq in ColabFold [57,58]. However, no co-factors such as heme were present in the structure modeled by AlphaFold2. To investigate the location of the heme in F6H4, we applied the predicted structure of apo-GmF6H to AlphaFill [59]

### 4.9. Statistical Analysis

We constructed a linkage map using AntMap Ver. 1.1 software [60] with default parameters. We identified QTLs by single-interval mapping using R/qtl software [61]. We used linear regression and analysis of variance (ANOVA) to confirm the effects of QTLs and to evaluate the association between the genotype of the closest DNA marker and isoflavone content. All analyses were performed using R software [62].

## Figures and Tables

**Figure 1 plants-13-00156-f001:**
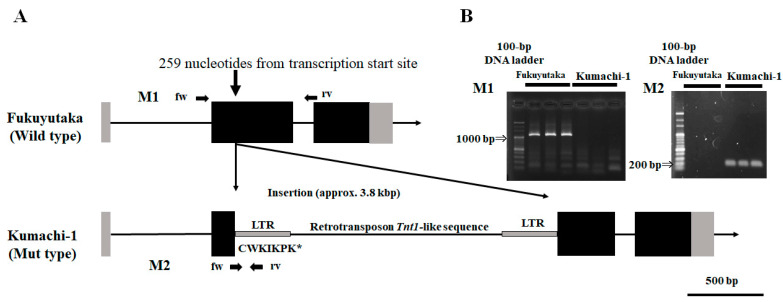
Gene structure comparison between Fukuyutaka (wild-type allele) and Kumachi-1 (null allele: Mut type). (**A**) The gene structure of *Glyma.11G108300* is shown with exons (black boxes), introns (lines), and 5′ and 3′ untranslated regions (gray boxes). The Kumachi-1 sequence contains an insertion of a *Tnt1*-like sequence with a long terminal repeat (LTR: narrow gray boxes). The physical locations of the primer pairs (M1 and M2) are shown. The truncated Mut-type amino acid sequence in Kumachi-1 allele with the insertion of Tnt1-like sequence is shown in a one-letter code with stop codon (asterisk). (**B**) Agarose gel electrophoresis of PCR products amplified with the M1 and M2 primer pairs. The far-left lanes contain a 100 bp size marker.

**Figure 2 plants-13-00156-f002:**
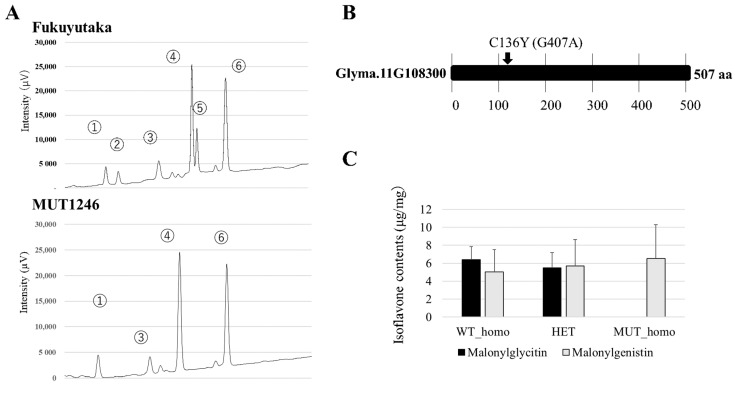
Identification of novel glycitein isoflavone-deficient mutants and the responsible genes. (**A**) Representative chromatograms of wild-type and MUT1246 mutant lines, with the latter showing a null phenotype for glycitein isoflavones (peak 2 and 5), as in Kumachi-1. The number of peaks are as follow, 1: daidzein, 2: glycitin, 3: genistin, 4: malonyldaidzin, 5: malonylglycition, 6: malonylgenistin. (**B**) Predicted C136Y amino acid substitution in Glyma.11G108300 (F6H4) of MUT1246 and the causal SNP with its position from the start codon of the gene. (**C**) Mapping experiment using a cross between MUT1246 and TOYO showing co-segregation between the SNP genotype of G407A (resulting in amino acid substitution by C136Y) and the glycitein isoflavone phenotype. The mutant homozygous genotype (MUT_homo) showed complete absence of malonylglycitin in the hypocotyls.

**Figure 3 plants-13-00156-f003:**
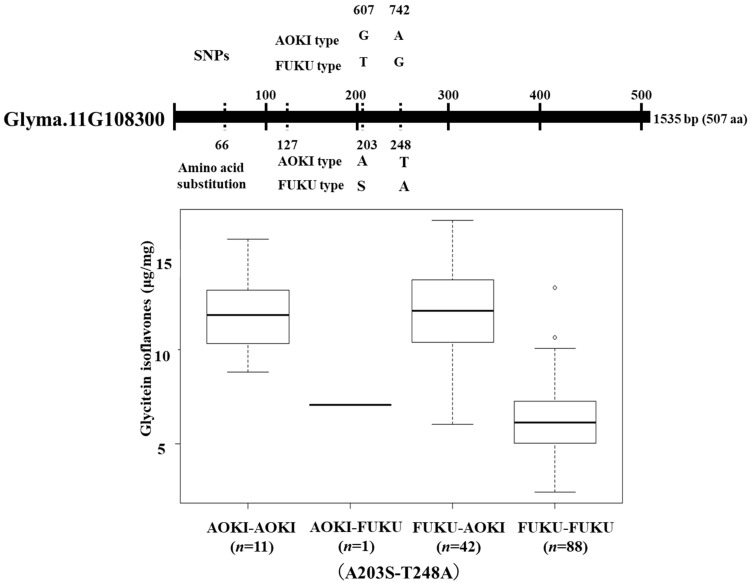
Interactive effect between SNPs in Glyma.11G108300 identified in an association study using the soybean mini-core collection. The positions of the SNPs causing amino acid substitutions are shown above the gene map. Dotted vertical lines show positions of SNPs between AOKI and FUKU. Non-synonymous amino acid substitutions are shown below the gene structure. Box plots show the amount of glycitein isoflavones produced by each genotype based on the amino acid substitutions encoded by Glyma.11G108300. Genotypes AOK-AOK, AOK-FUKU, FUKU-AOK, and FUKU-FUKU encode amino acid residues 203A and 248T, 203A and 248A, 203S and 248T, and 203S and 248A, respectively. The number of individuals with each genotype is shown in parentheses.

**Figure 4 plants-13-00156-f004:**
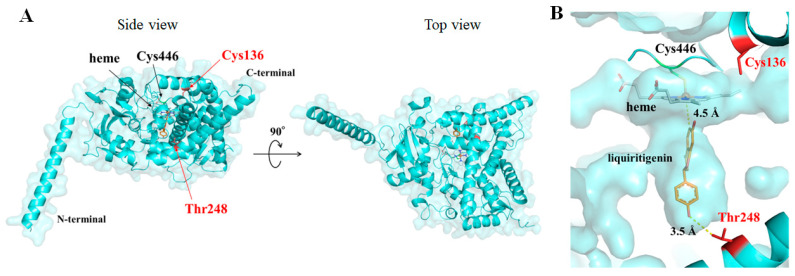
Modeled protein structure of F6H4 (encoded by Glyma.11G108300) with two amino acid substitutions causing null (136Y) and active (248T) proteins. (**A**) Overall structure of F6H4 (Aokimame type). The protein structure was modeled using AlphaFold2 and the position of heme was predicted using the AlphaFill program. (**B**) Active-site pocket of the modeled structure. The position of the substrate was provided by AlphaFill (Appendix A). The substrate for F6H4, liquiritigenin, was placed at the same position as the substrate (4-phenyl-1H-imidazole) in the modeled structure. The distances between 6C of the substrate and heme iron, and between 4′-OH of the substrate and 3-OH of threonine, were 4.5 and 3.5 Å, respectively. The heme, liquiritigenin, and amino acid side chains are shown as sticks. The iron ions are shown as spheres. The carbon atoms of F6H4, Cys446, heme, and liquiritigenin are colored as aquamarine, green, gray, and orange, respectively. Amino acids Cys136 and Thr248 are colored in red.

**Figure 5 plants-13-00156-f005:**
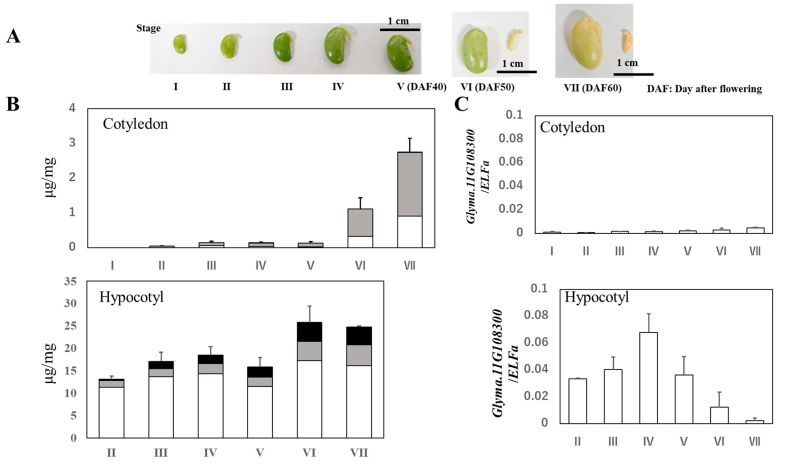
Accumulation of glycitein isoflavones during soybean seed development and tissue-specific expression of *Glyma.11G108300*. (**A**) Representative staging of seeds used for isoflavone extraction and expression analysis. Hypocotyl parts (plumule, epicotyl, hypocotyl, and radicle) were excised from the cotyledons, except at stage I, and analyzed separately. Separated hypocotyls are shown for stages VI and VII. (**B**) Isoflavone accumulation patterns in cotyledons and hypocotyls during soybean seed development. Each color indicates daidzein (white), genistein (gray), and glycitein isoflavones (black). (**C**) *Glyma.11G108300* expression profile during soybean seed development. Relative expression of *Glyma.11G108300/ELFa* was evaluated with standard deviation.

**Figure 6 plants-13-00156-f006:**
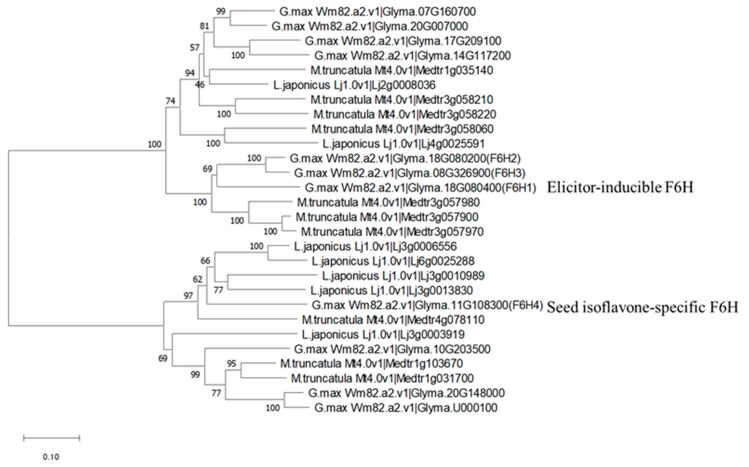
Phylogenetic tree of homologous amino acids sequences for flavonoid 6-hydroxylase in soybean and two other legumes, Lotus japonicus and Medicago truncatula. The MEGA-X program with the neighbor-joining method were used for the analysis.

**Figure 7 plants-13-00156-f007:**
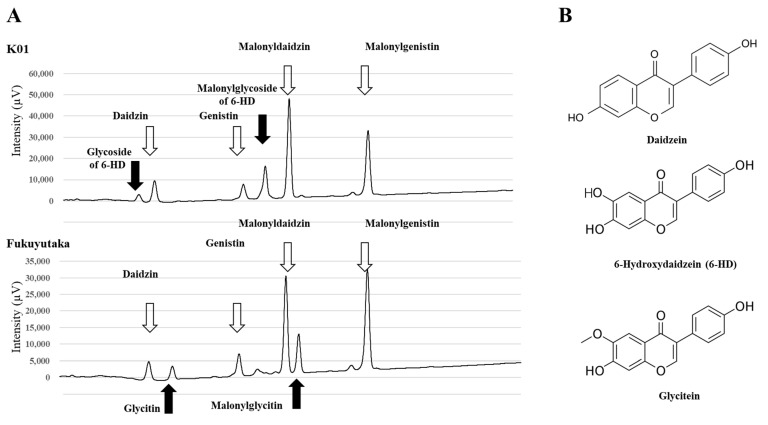
Identification of a novel glycitein isoflavone peak-shift mutant. (**A**) Representative HPLC chromatograms of isoflavones extracted from hypocotyl tissue of the mutant line (K01) and wild type (Fukuyutaka). Peaks in common (white arrows) and differing between the two lines (black arrows) are shown. (**B**) Chemical structures of daidzein, 6-hydroxydaidzein, and glycitein.

**Figure 8 plants-13-00156-f008:**
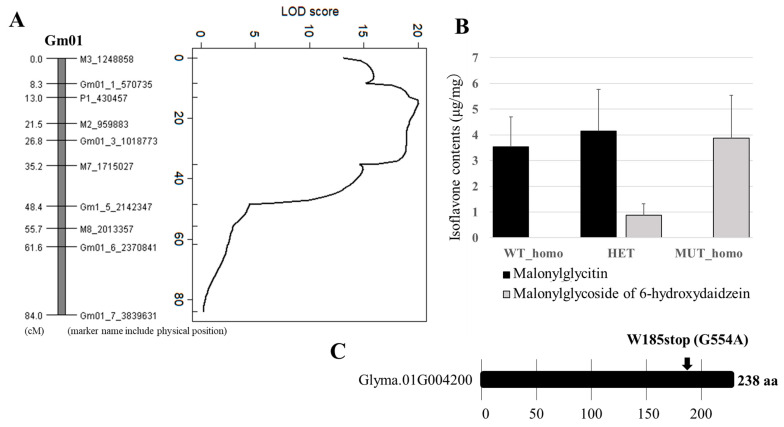
Mapping of the K01 mutated gene and location of premature stop codon. (**A**) Linkage map and QTL analysis of malonylglycitin content in a mapping population derived from Toyoshirome and K01. (**B**) Mapping experiments showing co-segregation between SNPs detected in Glyma.01G004200, malonylglycitin (black), and malonylglycoside forms of 6-hydroxydaidzein. (gray). (**C**) The predicted length of the peptide encoded by Glyma.01G004200 and the position of the truncation in K01.

**Figure 9 plants-13-00156-f009:**
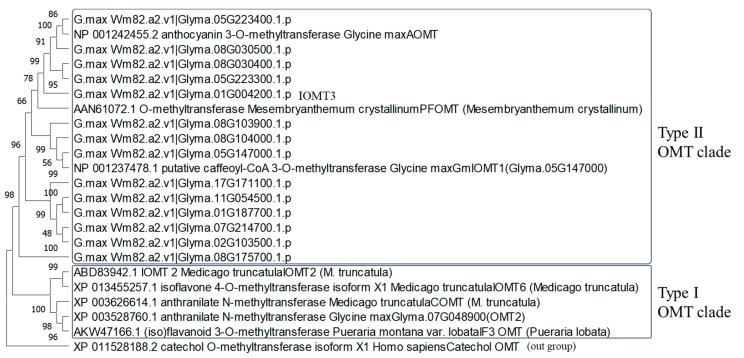
Phylogenetic tree of homologous amino acid sequences for isoflavone O-methyl transferase in soybean and other legumes. The MEGA-X program with the neighbor-joining method were used for the analysis.

**Figure 10 plants-13-00156-f010:**
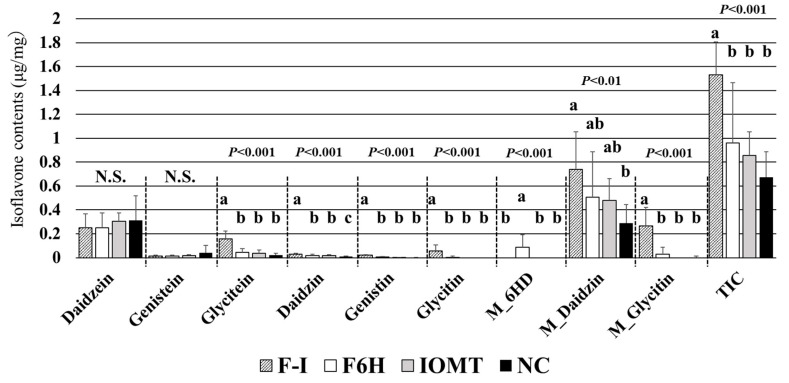
Isoflavone accumulation in transgenic soybean hairy roots with four constructs. F-I, F6H4, and IOMT3; F6H and F6H4 only; IOMT and IOMT3 only; NC, empty vector. Ten transgenic hairy roots for each construct showing high isoflavone content were used for this analysis. M_6HD, malonylglycoside forms of 6-hydroxydaidzein; M_Daidzin, malonyldaidzin; M_Glycitin, malonylglycitin; TIC, total isoflavone content. Significant differences with *p*-values by the Tukey–Kramer method are shown using lowercase letters; values marked with the same letter are not significantly different. N.S., no statistically significant differences for the indicated isoflavone.

## Data Availability

Data are contained within the article.

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
