# Peer review of "Identification of Genes Responsible for the Synthesis of Glycitein Isoflavones in Soybean Seeds"

_plants, 2024, doi:10.3390/plants13020156_

Round 1

Reviewer 1 Report

Comments and Suggestions for Authors

 The major concerns are as follows:

1. In the results, how to find the genes related to glycitein content in soybean seeds, which is not clarified in th beginning results. It suggests to introduce the details how to get the genes.  We don't know if there are another genes responsible to isoflavone content.

2. As we known that the key enzymes of F6H and IOMT are mainly responsible to isoflavone accumulation in soybean seeds.  These gene functions were primary clear based on the isoflavone pathway. Therefore, the innovation of these genes are not enough. 

3. In the Fig 2, 3, 7, there are not units in the axis of some HPLC profiles, which should be have a details.

4. As we known that the isoflavone content were different in the various parts of seeds. So that there should be presented in the details in the Figs.

5. Isoflavone content was significantly affected by genetic and enviromental factors.  So that the phenotypic data should be obtained in various enviromental plots. However, in this study the phenotypic data of isoflavone content were only determined on one environmental plot, which was difficult to get the precise data to validate the mapping results. How to avoid the situation.

6. In the reference,  the format of reference was not follow the standard of MDPI. Please check and revise them.

Comments on the Quality of English Language

The manuscript mainly introduced the genes related to glycitein content in soybean seeds, which can benefit to soybean isoflavone researchers. Howerver, in the results, how to find the genes, which is not in details. Therefore, it is suggested to revise the manuscript to clarify the major concerns. 

Author Response

Please see attachment file for the response to your comments.

Reviewer 2 Report

Comments and Suggestions for Authors

The report on identifying genes responsible for synthesizing glycitein isoflavones in soybean seeds was well written, and the experimental data support the results. Two F2 mapping populations were used to identify genes, and the gene structure and functions were confirmed.  The protein structure of F6H4 was modeled using AlphaFold2, and expression levels of F6H4 during soybean seed development were compared. The depth and contents of this study are excellent and ready for publication. But the title of the report is "Identification of Genes..." there should be some results to show how much phenotype variation of the gene can be explained from the 2 F2 mapping population so that readers know R2 of the genes, which breeders can use these genes/markers to do marker-assisted breeding for Glycitein Isoflavones. 

Author Response

Please see the attachment file for the responces to your comments.

Round 2

Reviewer 1 Report

Comments and Suggestions for Authors

The manuscript was revised based on the comments, which can be published in the Plants. However, there are 17 genes located in the QTL region qMgly-11, how to focus on the  gene Glyma.11G108300, which should have a explanation. In Fig5-B, what means in different colors legends. 

Comments on the Quality of English Language

The manuscript should be revised by the native English speaker

Author Response

Dear reviewer,

We are grateful for your comments and careful reading. We have improved our manuscript based on your comments.

I rewrote the reason for focusing on Glyma.11G108300 as follows:

(before revision)

“Among these genes, we considered Glyma.11G108300 (8.25Mbp on Gm11) annotated as p450, showing homology with other F6H proteins as a candidate of qMgly-11.

(After revision)

“We examined the annotation information of these genes and Glyma.11G108300 (8.25Mbp on Gm11) annotated as p450, showed homology with other F6H proteins. We considered this gene is reasonable for a candidate of qMgly-11.”

We have also added an explanation of the color of Fig5B as below.

“(B) Isoflavone accumulation patterns in cotyledons and hypocotyls during soybean seed development. Each color indicates daidzein (white), genistein (gray), and glycitein isoflavones (black).”

To ensure English language quality, I promise to undergo an English review before submitting the final version of the manuscript. Plants Editor, Ms. Li, requested us to submit the revised manuscript within three days.

Sincerely,

Satoshi Watanabe